# "Not Our Target Users": UX Professionals' Perceptions of Designing for Older Adults

Humaira Tasnim[*]

Faculty of Information, University of Toronto

Cosmin Munteanu[†]

Department of Systems Design Engineering, University of Waterloo

## ABSTRACT

In this paper, we revisit Jonathan Lazar's early work [24] on understanding designers' perceptions of accessibility for people with disabilities and follow the same approach to instead contribute similar insights into the current state of designing websites and web applications for seniors. For this, we present a survey investigating how design professionals consider digital accessibility and usability for the ageing population in the UX practice. The survey probed on awareness and application of usability principles for older adults and challenges that hinder the design of senior-friendly products. Findings reveal that many respondents did not incorporate senior-focused usability practices in their work, nor were they familiar with design principles specific to older users. Lack of awareness and knowledge regarding the accessibility and usability needs of older adults were stated to be the main barriers to senior-friendly design. The study identifies several other challenges facing UX professionals when designing for seniors and provides directions for future research.

**Keywords:** Older adults, inclusive design, UX professionals, user interface design, senior-friendly design guidelines.

## 1 INTRODUCTION

In recent years, with the push towards a more inclusive society, there has been an increasing demand for the consideration of diverse user profiles in the design of digital products and services. One such user profile is that of 'older adults,' an important and expanding group of internet users who are typically underrepresented in technology design. According to the United Nations [42], the global population aged 65 years or over is growing faster than all other age groups. With the unparalleled growth of the ageing population, the number of older adults using online technologies also continues to increase. Internet use doubled from 35% to 75% among seniors in the United States between 2007 and 2021 [35], and similar trends are occurring all around the developed world [14][32][39].

Despite their increasing technology adoption, older adults still struggle to use many online services due to various factors associated with ageing. As people age, they experience limitations in their functional abilities and are gradually afflicted with difficulties in vision, hearing, cognition, and mobility [10][26]. Generally, the user interfaces of many online products do not take these changing abilities into account, nor do they address the specific design needs of older adults [2][5][15][29], such as using legible fonts, larger text sizes, and color contrast for higher on-screen readability, or ensuring simplified layouts, familiar patterns, and clear feedback to avoid memory-related issues. This compromises their usability and causes increased frustration for senior users, resulting in a lack of self-confidence and motivation to continue using the technology [43]. Older adults' mental models of user interfaces are also often different compared to those of (younger) designers who design these products and services [1]. This puts seniors at a major disadvantage in the digital age as they are unable to access the same services as their younger counterparts. Previous studies confirm that a more accessible web can be instrumental in enabling older adults to maintain an active, community-based lifestyle [8][9][18]. Therefore, to facilitate their ageing independently and to provide them equal access to information, technology designers should take into account the needs of the ageing population and ensure the products they design are "senior-friendly," i.e. easy for seniors to use without any additional help.

Usability for older adults has been a growing topic of interest and relevance for the human-computer interaction (HCI) community over the years. Studies have provided literature reviews [6], conducted expert reviews of websites [7], and discussed methodologies of user-centered design through participatory design [12] or usability testing [30] with seniors. Several design guidelines [23] and heuristics [26] have also been published to assist in improving the usability of interfaces for older adults. However, there remains a lack of research on how user experience (UX) professionals in the industry approach this topic. With the large number of online services that lack usability for older adults [2][5][15][29], it is important to assess where the design community currently stands in terms of senior-focused design practices. This also raises the need to identify any barriers UX professionals might be facing that are inhibiting the design of user-friendly interfaces for older adults. Prior research has extensively investigated such barriers to (and attitudes toward) the widespread use of accessibility design guidelines, including seminal work on which we methodologically ground our own [24]. However, accessibility guidelines may not provide comprehensive support when designing for older adults [27][38]. In fact, there are also strong arguments against equating ageing with accessibility (in design and elsewhere) [21][28][36].

Therefore, to fill this gap, we conducted an online survey with the participation of 130 professionals working in UX design from various industries. The aim of this study was to:

1. investigate the level of understanding and awareness UX professionals have about accessibility and usability for seniors,
2. understand how UX professionals incorporate accessibility and usability for seniors in their design projects, and
3. uncover the motivations for, barriers to, and challenges UX professionals face when ensuring senior-focused design usability.

This paper presents the findings of the study in detail and provides directions for future research. As a note on terminology, the terms 'older adults' and 'seniors' have been interchangeably used in this paper. While the term 'older adult' is more commonly used in HCI literature, the more prevalent term in our own socio-cultural context is that of 'senior' (as indicated by government

---

[*] humaira.tasnim@mail.utoronto.ca
[†] cosmin.munteanu@uwaterloo.ca

surveys in our geographical location, in a large urban center in Canada).

This study makes a major contribution to research on UX professionals by providing an insight into their current state of awareness and their application of design methodologies for older adults. This is noteworthy because, while accessibility practices for people with disabilities have been widely studied, there has been limited focus on professionals' expertise and experience with designing specifically for seniors. Through this research, we bring to surface evidence about several reasons why senior-friendliness is not a focus for UX professionals while fostering a reflection on the transfer of research-based recommendations to the professional environment. The results provide insights into the current resources and attitudes designers have with regards to designing for seniors in comparison to previous studies which have focused on different, yet related domains (e.g. accessibility).

While we were inspired by prior work on understanding the challenges designers face when designing for accessibility, our work is not about accessibility. Instead, our study only draws methodologically from Jonathan Lazar's prior work on perceptions of accessibility [24]. We adapt Lazar et al.'s approach and extend the scope of their methods to studying the challenges designers have with respect to usability for seniors. Our findings reveal that similar awareness (and work) is now needed in the field of senior-friendly design as it was for accessibility at the time of Lazar et al.'s seminal paper, and through this, we hope that the results of our study will inspire a culture and policy shift in terms of including older adults in design as Lazar et al.'s paper did for accessible design.

## 2 BACKGROUND AND RELATED WORK

This section provides the theoretical background of the study and situates our work in literature. We begin with an overview of how digital accessibility and usability are relevant to designing for seniors, followed by a description of various design principles and usability methods available to assist UX professionals in the creation of senior-friendly products. We conclude this section with a discussion of how lack of specific support resources within industry may result in designers not being aware or knowledgeable of ways to make their products inclusive – this was revealed by Lazar et al. [24] in their seminal work related to designing for accessibility, which we now aim to replicate with respect to designing for older adults.

### 2.1 Digital Accessibility and Seniors

The two important concepts when designing interfaces that are inclusive of older adults' needs and limitations are 'digital accessibility' and 'usability'. While these are two distinct aspects, they are closely inter-connected in the context of crafting technologies that work for everyone.

*Digital accessibility* primarily focuses on people with disabilities and ensures that technologies are designed and developed in a way that everyone can use them, regardless of disability type or severity of impairment [44]. This includes auditory, cognitive, neurological, physical, speech, and visual impairments that may affect people's access to, or interaction with online products and services. While digital accessibility predominantly serves people with disabilities irrespective of age, it also benefits people *without* disabilities, like older adults who sometimes face gradual limitation of functional abilities due to ageing [47]. For example, one of the accessibility principles entails allowing users to incrementally change the text size in user interfaces. Although this principle is targeted at people with disabilities, senior users requiring larger text in interfaces due to declining vision can also gain from its implementation. Older adults can therefore be assumed to be beneficiaries of accessible design depending on their own personal circumstances, which makes accessibility an important consideration for UX professionals when designing digital products.

In many countries, accessibility of digital designs is now legislated, and it follows from widely used industry standards such as the Web Content Accessibility Guidelines (WCAG), published by the World Wide Web Consortium (W3C) Web Accessibility Initiative (WAI) [45]. These guidelines have become the benchmark for creating and evaluating accessible interfaces and have been set as the minimum requirement in the digital accessibility policy of many countries worldwide [37]. While WCAG has been primarily developed for websites, the success criteria for these guidelines are not technology-specific and, therefore, they apply to all kinds of user interfaces. It is important to note that these guidelines are highly technical and require expert knowledge of web technologies for their comprehension and application [22]. However, there are a variety of software tools available that complement these guidelines and can help professionals determine if their design meets accessibility standards [46].

*Usability*, on the other hand, refers to the general intuitiveness and ease of use of user interfaces. Usability for seniors ensures that digital products can be used by older adults to achieve their goals in an effective, efficient, and satisfactory manner, and the level of usability is determined by how well the features of the user interface accommodate senior users' needs and contexts [3]. Various senior-focused usability guidelines have been formulated over the years by researchers to help professionals design for older adults (Section 2.2).

Projects such as the Web Accessibility Initiative: Ageing Education and Harmonization (WAI-AGE) [47] suggest following accessibility guidelines to remove some barriers for older adults, however this often only covers the most basic aspects of how older adults engage with digital designs, leading them to be unsure of their ability and unmotivated to continue trying new technologies [43]. Significant research within HCI also highlight the dangers of conflating ageing with accessibility [21][28] – which we took significant care to avoid in our own research, especially as we are drawing methodologically from prior research on accessibility. Therefore, it is crucial to take into account usability guidelines that specifically cater to older adults (in addition to the general accessibility guidelines) during the design process.

### 2.2 Senior-friendly Design Guidelines

Previous studies show that usability is one of the most important factors affecting older adults' adoption of technology [25]. For example, in a study examining the usage of electronic personal health records [40], it was found that while seniors considered these systems to be valuable, the prevalence of usability problems, such as complex navigation systems and highly technical language, made them challenging to use by older adults. Lack of usability can also result in increased frustration for older adults. Therefore, in order to avoid usability challenges for older adults, it is important for UX professionals to adhere to a user-centered design approach and consider the needs and pain points of seniors in the design and evaluation of technologies.

Design principles explicitly targeted at the needs of older adults have been established to ensure the usability of interfaces for seniors. Based on various research conducted on ageing, the National Institute on Aging (NIA) and the National Library of Medicine (NLM) in the United States published "Making Your Web Site Senior Friendly: A Checklist," consisting of design guidelines that are very specific to older adults [31]. Examples of guidelines from the checklist include providing clear instructions, avoiding jargon, making it easy for users to enlarge text, reducing

scrolling, and using high-contrast color combinations. Similarly, Kurniawan and Zaphiris [23] presented a set of "research-derived ageing-centered web design guidelines" for older adults in 2005, which include providing larger targets, having clear navigation, using color and graphics minimally, and reducing demand on the users' memory. In 2013, Lynch, Schwerha, and Johanson [26] developed a weighted heuristic for evaluating the usability of user interfaces for older adults, which included a list of 32 characteristics representing the most important senior-friendly design recommendations. The Nielsen Norman Group also released their third edition of "UX Design for Seniors (Ages 65 and older)" in 2019, which is a commercially available report outlining design guidelines for particular tasks and web components to support usability for seniors [33]. Although these guidelines have been widely used and referenced in academia, there is limited research on how much of these recommendations are transferred to the professional environment and how UX professionals incorporate them into their design practice.

### 2.3 Involving Seniors in the Design Process

While following both accessibility guidelines and usability principles are important, they are not sufficient to guide designers toward senior-friendly design. To test the effectiveness of these guidelines and to ensure all needs and pain points of older adults are taken into consideration, senior users should be directly involved in the design process through various usability methods. Yesilada et al. [48] found that designers believe accessibility evaluation should be grounded on user-centered design practices, as opposed to just inspecting source codes, in order to obtain more reliable and valid results. This sentiment was also shared by Hart, Chaparro, and Halcomb [17], who suggested using a combination of design guidelines and usability testing when designing websites for older adults, as well as Milne et al. [27], who recommended designers go beyond WCAG and get first hand interaction with users to ensure their needs are met.

### 2.4 Understanding Designers' Attitudes and Barriers toward Inclusive Design

Design professionals from various interdisciplinary backgrounds participate in the design and development of online products and services, and therefore, their perceptions and practices of accessibility have been an important topic investigated by several research projects. Five relevant surveys conducted with these professionals are summarized below:

Lazar, Dudley-Sponaugle, and Greenidge (2004) surveyed 175 webmasters of government and commercial organizations to investigate their knowledge of web accessibility [24]. Most of the participants (74%) reported that they were familiar with government laws on web accessibility, and many (79%) were familiar with automated software tools used for accessibility evaluation. These results indicate that a lack of knowledge or awareness is not the prime reason behind the shortage of accessible interfaces. However, it is also notable that almost one-fourth of the respondents (23%) did not know about web accessibility guidelines at all. Participants cited lack of time, training, managerial and client support, as well as lack of software tools, and confusing accessibility guidelines as the main barriers to web accessibility. They also mentioned concerns regarding maintaining a balance between accessibility and good graphic design, which appears to stem from the misconception that an accessible website may downgrade the experience for visual users [11]. Concerning motivation, participants indicated that the primary reasons for making their websites accessible would be requirements imposed by the government, use of the websites by people with disabilities, external funding, requirements from

management or clients, training on accessibility, and access to better accessibility tools.

A similar survey was conducted by ENABLED Group (2005) with 269 subjects, which included webmasters, managers, and content editors [13]. Only 36% of the participants responded that they try to make their websites accessible, and very few (13%) had received training on accessibility. The primary reasons behind this were indicated to be a lack of knowledge of web accessibility guidelines, lack of technical knowledge, and time constraints. Nonetheless, many participants (74%) expressed interest in attending training sessions to learn more about accessibility, with the preferred topics being web accessibility guidelines, usability, and accessibility evaluation.

Freire, Russo, and Fortes (2008) surveyed 613 professionals in Brazil from diverse backgrounds (academia, industry, and government), who took part in web development projects [16]. The findings showed that only 20% of the participants considered accessibility as critical to their projects. Lack of training on accessibility and lack of knowledge about the Brazilian accessibility law were stated to be the primary reasons behind accessibility not being a priority among participants.

Modelling the studies mentioned above, Inal, Rızvanoğlu, and Yesilada (2019) surveyed 113 UX professionals in Turkey regarding their awareness and practice of web accessibility [19]. While most participants (71%) indicated that they had received training on web accessibility, many (69%) still did not consider accessibility in their projects. Moreover, only 17% of the participants reported working directly with people with disabilities for their projects and accessibility evaluations. A similar survey was conducted by Inal et al. (2020) with the participation of 167 UX professionals from Nordic countries [20]. Results show that while digital accessibility was considered to be important by the respondents, they had limited knowledge about accessibility guidelines and standards. Most of the organizations represented in this study included accessibility in their projects, however, the time spent by these organizations on accessibility issues was reported to be very less. The main challenges participants faced in creating accessible systems were lack of training and time and budget constraints.

In summary, the studies conducted by the ENABLED Group [13] and Freire et al. [16] confirm that a lack of awareness of accessibility laws and a lack of training on web accessibility can largely hinder the development of accessible interfaces. On the other hand, the studies conducted by Lazar et al. [24] and Inal et al. [19] show that awareness or knowledge about web accessibility does not automatically lead to the development of accessible interfaces. Although design professionals are aware of the needs of people with disabilities, they still do not take these needs into consideration generally.

**The above-mentioned studies are mostly centered around accessibility for people with disabilities, which is different from that for seniors, as discussed earlier.** While WCAG guidelines can be applicable to older people experiencing age-related impairments [47], merely following accessibility guidelines does not necessarily lead to the design being usable, nor do they help overcome the particular challenges facing older adults [27][38]. There is still much work to be done in ensuring usability for seniors, as can be understood from the results of numerous previous studies [2][5][15][29] which revealed how websites or apps are lacking in this regard. It has also been identified that there appears to be little awareness among designers of the specific requirements of older people compared to their knowledge of WCAG [38]. As a result, they are not considering the particular needs of a growing audience when designing user interfaces.

# 3 STUDY RATIONALE AND METHODS

Based on the surveyed literature, **we claim that designing digital applications for older adults is today struggling with similar challenges as designing for accessibility did more than two decades ago**. As such, it is imperative to find the reasons behind the lack of senior-friendly interfaces and to fill this gap, research concerning the perceptions of UX professionals in considering accessibility and usability for older adults needs to be done. In this vein, we are inspired by Lazar et al.'s [24] landmark research on web accessibility. We draw methodologically from that seminal research that expose gaps in the design process with respect to accessibility.

Aiming to extend Lazar et al.'s [24] work on digital accessibility (by extending this to usability for seniors), we methodologically followed their protocol and adapted it to the emerging context of inclusive design for older adults. As such, we employed a quantitative survey-based methodology for this study. The survey was administered online using SurveyGizmo with 130 respondents. The survey was deployed in 2019, with the bulk of data collection occurring throughout 2020 (with several interruptions due to COVID-19 pandemic's effect on availability of research staff – however we do not consider this extended period of recruitment to have any influence on the quality of survey responses since no time-sensitive information was collected.)

## 3.1 Research Questions

Through our research, we attempt to address the identified gaps in the literature by focusing on three main research questions:

RQ1: What is the level of understanding and awareness UX professionals have about accessibility and usability for seniors?

RQ2: How do UX professionals incorporate accessibility and usability for seniors in their design projects?

RQ3: What are the motivations for and challenges of ensuring usability for seniors by UX professionals?

## 3.2 Questionnaire Design

The questionnaire was derived from priorly validated research instruments on digital accessibility awareness and practices (see 3.2.1). We opted for this approach due to our assumption that designing for seniors may be at the same stage of awareness and practice as designing for accessibility was when Lazar et al. [24] conducted their seminal research on this topic. Additionally, using a priorly validated instrument (questionnaire) that was used in a similar domain facilitated the collection of more robust data which may not have been possible with an instrument developed from the ground up.

The questionnaire was subject to two rounds of pilot testing. The first round was conducted with two participants from academia and one participant from the industry. Questions were revised to address issues of clarity and ambiguity that emerged from the pilot. For the second round, the questionnaire was deployed online via SurveyGizmo, and was validated by two participants from academia. The final questionnaire was comprised of 32 questions, both open-ended and closed-ended, grouped into four sections:

1. *Personal Information* included eight questions to obtain demographic information, such as geographic location, educational background, and work experience;
2. *General Understanding and Awareness* included nine questions pertaining to RQ1, to determine knowledge of how seniors use the web, and awareness of assistive technologies, digital accessibility legislation, senior-friendly design guidelines, and tools;

3. *Practical Experience* included ten questions pertaining to RQ2, to identify consideration of accessibility and usability for seniors in the UX practice, and the use of various research methods and evaluation techniques;
4. *Motivations and Challenges* included five questions pertaining to RQ3, to understand challenges, and personal and organizational interests in supporting usability for seniors.

We clarify here that questions in the *General Understanding and Awareness* and *Practical Experiences* groups were designed to compare 'general accessibility' practices with 'usability for seniors' practices. Questions in Motivations and Challenges focused only on 'usability for seniors'. Since our focus was on attitudes towards designing for seniors in general, we did not hypothesize anything specific about accessibility and usability. As such, the Results section is presented from the responses that emerged from these questions, and not from a preconceived structure.

The questionnaire was preceded by a consent form that outlined the purpose of the study, explained the rights of the participants, and assured them of complete anonymity. Following the consent form, participants were taken to a separate web page where they were presented with the questionnaire.

### 3.2.1 Grounding of Questionnaire Design in Prior Work

Most questions were closely informed by previously developed and validated surveys, such as Lazar et al. [24] and Freire et al. [16], and extended to inquire about "designing for seniors", as opposed to "designing for people with disabilities". A breakdown of the survey questions by the source is provided in Table 1. The survey instruments are entirely available as supplementary materials included with the submission of this paper. The questions we included from Lazar et al.'s and Freire et al.'s instruments were selected based on how applicable these were to the process of considering various resources (e.g. guidelines) in making designs inclusive to a specific user group that was typically excluded from design considerations. This has allowed us to easily and objectively adapt their instruments (which were focused on accessibility) to our own domain – designing for older adults.

Table 1: Number of survey questions by source

| Survey Section | No. of questions derived from Lazar et al. [24] | No. of questions derived from Freire et al. [16] | No. of questions added by authors |
|---|---|---|---|
| General Understanding and Awareness | 3 | 4 | 2 |
| Practical Experiences | 2 | 2 | 6 |
| Motivations and Challenges | 3 | 1 | 1 |

The extra questions we added were also extended versions of questions from Lazar et al. [24] and Freire et al. [16]. These questions were included to help gather data on senior-friendly design practices, which was not addressed in their study. For example, Lazar et al.'s question: "Are you familiar with any of the following accessibility guidelines from the Web Accessibility Initiative?" was extended to "Are you familiar with the senior-friendly design guidelines published by the National Institute on Aging and the National Library of Medicine?", while the question: "What do you think is the biggest challenge of making a website accessible for users with visual impairments?" was modified to "What do you think are the challenges of making

websites or apps senior-friendly?". Questions regarding visual impairments were asked to examine general accessibility practices (similar to Lazar et al.), which can be a part of the UX practitioner's design process when designing for seniors. For example, the question: "Have you ever created a website that is accessible for users with visual impairments?" was extended to "Have you ever created a website or app that is accessible for seniors?". Similarly as an example, Freire et al.'s survey question asking the respondent to describe "Awareness of problems faced by blind people using the Internet" was adapted to our domain as "Describe your understanding of how seniors use websites". A complete description of our survey, including how questions were derived from the instruments used in Lazar et al.'s and Freire et al.'s research and adapted to our domain (older adults), is included in the supplementary materials submitted together with this paper.

Deriving our questionnaire from previously-validated instruments required us to compress some questions and also not inquire about accessibility practices at the same level of granularity. As such, while questions about familiarity with specific accessibility tools were not included, an option was provided for participants to type in the tools they were familiar with. However, given the relevance to designing for older adults [15], we included questions regarding visual impairments, which were asked to examine general accessibility practices (similar to Lazar et al.), which can be a part of the UX practitioner's design process when designing for seniors. Questions regarding assistive technologies were adopted from Freire et al. with very minor modifications, and participants were asked to choose from a very broad range of answers. The same set of options was also used by Inal et al. [19]. In the same manner, the question about ethical consideration was not completely removed; it was included as part of the motivation-related questions.

### 3.2.2 Definition of 'Seniors'

Studies in literature vary in their definition of 'seniors' and the age range they belong to. Generally, the age range defined for 'seniors' is either over 60 or over 65. However, some research uses a lower threshold or a flexible threshold by tying it to the typical or legal retirement age. Given that the participants in this study were not older adults, but rather UX professionals of varying ages, it was considered that imposing a standard age range for seniors may have been limiting and also insensitive to the localized and personal socio-cultural norms in which each UX professional may operate. Therefore, when answering the questions, participants were asked to think of their own definition of 'seniors' and the age range they belonged to as relevant to their culture and experiences.

### 3.3 Recruitment

Prospective participants were invited to express interest in the study by filling out a short enrollment form which served as a screener to ensure quality of data and to avoid fraudulent responses [41]. The enrollment form was posted on professional UX design groups on various closed-group (member-only) social media channels and promoted through personal contacts and announcements posted on e-newsletters and social media groups informally associated with several design communities. Participants were asked to briefly describe their work experience in the enrollment form, and those considered to be 'legitimate' responders [41] with a background in UX, were emailed a link to the questionnaire.

Participation was not restricted to a geographical location, given that this was an online survey. As a token of appreciation for their time and contributions, participants were offered a $10 Amazon gift card in a currency of their choice (US dollars or Canadian dollars) once they signed the consent form.

### 3.4 Participants

In total, 130 participants completed the survey. Participants had to meet the following eligibility criteria: be at least 18 years of age and be a design professional.

We used the term 'design professional' in the survey instead of 'UX professional' to include people who did not have UX-specific job titles but were still involved in user-centered design processes. For the purpose of this study, a 'design professional' is defined as someone who designs or provides consultancy services in the design of user interfaces for websites or apps that are not for their own personal use. This clarifying definition was provided in the consent form to help prospective participants decide whether they identified as design professionals.

Since the survey did not ask for personal information, there was no risk to participants self-identifying as design professionals. Whether the participants actually worked as design professionals was not verified, since requiring formal verification of participants' professions was not in line with the ethical guidelines for requesting excessive personal data. This limitation was mitigated by the recruitment strategy, as the study was only advertised on closed professional UX groups.

### 3.5 Analysis

Data obtained from the online survey was exported from SurveyGizmo and collated in a spreadsheet. The responses were then reviewed to ensure completion and consistency and to identify duplicates or outliers. Responses to open-ended questions were reviewed for quality by checking if the answers provided were relevant to the questions asked, in order to avoid any fraudulent responses [41].

Following quality assurance checks, data was then processed and coded to carry out the analysis. Descriptive statistics were employed to analyze the quantitative data provided by the multiple-choice questions. Comparative statistical analysis was not conducted due to the exploratory nature of the data and research questions.

Free text responses to open-ended questions were subjected to a thematic analysis to identify patterns across the data, and to complement and contextualize the quantitative findings from the survey. The thematic analysis was done using a data-driven inductive approach by the lead author — an experienced UX designer and techno-social researcher. Responses were systematically coded and reviewed for common, emergent themes following the guidelines by Braun and Clarke [4]. Since the responses were short and fact-oriented, and also not the main source of analysis for the predominantly quantitative instrument, a more extensive dual-annotator analysis was not necessary.

## 4 RESULTS

This section presents a synthesis of the data collected from the survey. Survey findings have been presented in tabular format to ensure accessibility (see supplementary materials for graphical format). Percentages have been rounded to the closest number in the text to improve readability.

### 4.1 Demographics

A summary of the demographic profile of the 130 participants is presented in Table 2. Participants worked in various industries, with the principal organizational areas being information technology (37%), followed by education (18%) and finance (11%). The job titles of the participants included a wide range of UX roles, such as UX designer, product designer, design lead, UX architect, UX researcher, chief design officer, design strategist, UI designer, information architect, and UX consultant. The average length of the participants' work experience as a

professional or a consultant in the field of UX was 6.55 years (SD=6.09). Regarding their personal rating of experience in the field, most of the participants described their level of experience as intermediate (44%) or advanced (32%). In terms of education, most of the participants had some form of post-secondary education with either a bachelor's degree (55%) or a master's degree (29%). A large number of participants (74%) also received professional training or education in the fields of UX and/or HCI.

Table 2: Demographic profile of participants (n=130)

| Variables | Responses | n | % |
|---|---|---|---|
| Geographic location | Canada | 55 | 42.3 |
| | United States | 41 | 31.5 |
| | Other | 34 | 26.2 |
| Industry | Education / Research | 23 | 17.7 |
| | Finance / Banking / Insurance | 14 | 10.8 |
| | Government / Military | 4 | 3.1 |
| | Healthcare / Medical | 3 | 2.3 |
| | Information Technology | 48 | 36.9 |
| | Telecommunications | 3 | 2.3 |
| | Other | 29 | 22.3 |
| | Not Sure | 6 | 4.6 |
| Education | High school degree or equivalent | 5 | 3.8 |
| | Some college, no degree | 11 | 8.5 |
| | Associate degree | 3 | 2.3 |
| | Bachelor's degree | 72 | 55.4 |
| | Master's degree | 37 | 28.5 |
| | Professional degree | 1 | 0.8 |
| | Doctorate | 1 | 0.8 |
| Professional education in HCI/UX | Yes | 96 | 73.8 |
| | No | 19 | 14.6 |
| | Other | 8 | 6.2 |
| | Not sure | 7 | 5.4 |
| Level of experience in HCI/UX | Expert | 19 | 14.6 |
| | Advanced | 42 | 32.3 |
| | Intermediate | 57 | 43.8 |
| | Basic | 12 | 9.2 |
| Web accessibility training/education | Undergraduate courses | 30 | 23.1 |
| | Graduate courses | 10 | 7.7 |
| | Online courses | 56 | 43.1 |
| | Training in the workplace (current or past) | 54 | 41.5 |
| | Other | 14 | 10.8 |
| | No training or education in web accessibility | 31 | 23.8 |

## 4.2 General Understanding and Awareness

### 4.2.1 Digital accessibility training and education

Participants were asked what kind of professional training or education they received in web accessibility through a multiple selection question. Most participants (76%) received some form of web accessibility education with the most common sources being online courses and workplace training programs, followed by undergraduate and graduate courses. Some of the other sources mentioned by participants include conferences, meetups, webinars, bootcamps, and personal research. It is notable that almost one-fourth of the participants did not receive any professional training or education in web accessibility (Table 2).

### 4.2.2 Understanding senior user needs

Concerning understanding of senior user needs, 58% of the participants stated that they knew how seniors use websites and how to design for them. The remaining 42% did not know how to design for seniors, and among them, 15% had no knowledge of how seniors use the web. Since many older adults use assistive technologies to access digital services, participants were also asked to specify all the assistive technologies they were familiar with through a multiple selection question. Almost all participants (96%) were familiar with assistive technologies, with the most popular selections being speech recognition tools, screen magnifiers, and screen readers (Table 3).

### 4.2.3 Web accessibility legislation and guidelines

Only 49% of the participants reported that they were familiar with government laws on digital accessibility. Responses from a follow-up question regarding the level of familiarity they had with the accessibility laws have been summarized in Table 3. 36% of the participants reported understanding and following digital accessibility laws. On the other hand, 47% of the participants barely knew or never heard of any accessibility laws.

Table 3: General understanding and awareness

| Responses | n | % |
|---|---|---|
| Level of understanding of how seniors use websites | | |
| I am aware that seniors can use websites, but I don't know how they use them | 19 | 14.6 |
| I know how seniors use websites, but I don't know how to design for them | 36 | 27.7 |
| I know how seniors use websites and how to design for them, but I haven't designed for them | 40 | 30.8 |
| I know how seniors use websites and I have designed for them | 35 | 26.9 |
| Familiarity with assistive technologies | | |
| Screen reader | 109 | 83.8 |
| Screen magnifier | 110 | 84.6 |
| Braille-based tools (e.g. printers, embossed printers) | 55 | 42.3 |
| Text-only browser | 74 | 56.9 |
| Alternative keyboard | 53 | 40.8 |
| Alternative mouse and joystick | 44 | 33.8 |
| Speech recognition tools (e.g. Siri) | 110 | 84.6 |
| Other | 4 | 3.1 |
| I am not familiar with any assistive technology | 5 | 3.8 |
| Level of familiarity with accessibility laws | | |
| I know the relevant law(s) and its web-related implications, and follow it | 47 | 36.2 |
| I know the relevant law(s) and its web-related implications, but don't follow it | 11 | 8.5 |
| I know the relevant law(s), but not its web-related implications | 11 | 8.5 |
| I have heard about it / I barely know about it | 29 | 22.3 |
| I have never heard about it | 32 | 24.6 |
| Familiarity with accessibility guidelines | | |
| Web Content Accessibility Guidelines (WCAG) | 88 | 67.7 |
| Authoring Tool Accessibility Guidelines (ATAG) | 8 | 6.2 |
| User Agent Accessibility Guidelines (UAAG) | 12 | 9.2 |
| I am not familiar with any accessibility guidelines | 39 | 30 |

Participants were asked which accessibility guidelines they were familiar with through a multiple selection question, and many reported being familiar with the WCAG (68%). It is also worth mentioning that 30% of the participants were not familiar with any accessibility guidelines. Regarding knowledge of accessibility checking tools, 64% reported being familiar with

these tools with specific mentions of WAVE, AChecker, Axe, Google Lighthouse, Contrast Analyzer, Siteimprove, etc.

### 4.2.4 Senior-friendly design guidelines

Most participants (83%) were not familiar with the senior-friendly design guidelines published by the NIA and NLM. Only 12 participants (9%) reported knowing about these guidelines. When inquired about familiarity with other senior-friendly design guidelines, most participants (73%) did not know of any other senior-friendly design guidelines either.

## 4.3 Practical Experiences

### 4.3.1 Web accessibility and usability for seniors as part of projects

Concerning previous experience, 54% of the participants reported to having designed accessible interfaces for users with visual impairments. On the other hand, 41% reported to never having created any website or app that was accessible to users with visual impairments. Likewise, in terms of designing for older adults, 43% of the participants previously created websites or apps that were accessible to seniors, while 40% had no previous experience designing for seniors.

Majority of the participants (81%, n=105) reported considering accessibility in the design projects they were involved in. Few of them (6%, n=8) did not consider accessibility in their projects and they were asked to explain their reasons for doing so through an open-ended question. All eight participants responded to the question and the reasons stated by them can be classified under the following themes: accessibility not being included in project scope (n=4), accessibility not being a requirement for the target group/customer (n=2), time and budget constraints (n=1), lack of client support (n=1), and lack of information and tools for accessibility (n=1).

In terms of senior-friendliness, only 40% (n=52) of the participants considered usability for seniors in the design projects they were involved in, while 34% (n=44) stated that they did not consider designing for seniors. The reasons given by 43 of these participants for not considering senior-friendliness in their projects can be classified under the following themes: senior-friendliness not being a requirement for the target group/customer (n=32), lack of awareness of how seniors use the Internet (n=4), senior-friendliness not required by clients/stakeholders (n=4), senior-friendliness not being a priority (n=3), no opportunity to interact with seniors (n=2), lack of knowledge about designing for seniors (n=1), limited project scope (n=1), and time and budget constraints (n=1).

### 4.3.2 Research methods for accessible and senior-friendly designs

The 105 participants who reported considering accessibility in their design projects were asked which research methods they used to design for users with disabilities through a multiple selection question. A similar question regarding research methods was asked to the 52 participants who reported considering senior-friendliness in their design projects. According to the responses, the most widely used method for both designing for users with disabilities and designing for seniors was following accessibility guidelines (Table 4).

### 4.3.3 Evaluation techniques for accessible and senior-friendly designs

The participants who reported considering accessibility and/or senior-friendliness in their design projects were also asked about the evaluation techniques they used when designing for people with disabilities or seniors. For designing for people with disabilities, the most prominent evaluation technique among the 105 participants was checking for compliance with accessibility guidelines. On the other hand, for evaluation techniques for designing for seniors, the most widely used technique among the 52 participants was conducting usability tests with seniors (Table 4).

Table 4: Use of research methods and evaluation techniques

| Research methods | Designing for users with disabilities (n=105) | | Designing for senior users (n=52) | |
|---|---|---|---|---|
| | n | % | n | % |
| Follow accessibility guidelines | 80 | 76.9 | 32 | 61.5 |
| Follow senior-friendly design guidelines | 20 | 19.2 | 12 | 23.1 |
| Conduct interviews | 34 | 32.7 | 16 | 30.8 |
| Conduct surveys | 24 | 23.1 | 12 | 23.1 |
| Generate personas | 32 | 30.8 | 20 | 38.5 |
| Conduct usability tests | 31 | 29.8 | 21 | 40.4 |
| Conduct participatory design sessions | 15 | 14.4 | 7 | 13.5 |
| Conduct heuristic evaluations | 48 | 46.2 | 20 | 38.5 |
| Other | 8 | 7.7 | 3 | 5.8 |
| I don't use any research methods | 8 | 7.7 | 6 | 11.5 |
| **Evaluation techniques** | | | | |
| Conduct usability tests with users with disabilities | 33 | 31.4 | 14 | 26.9 |
| Conduct usability tests with seniors | 35 | 33.3 | 28 | 53.8 |
| Test with automatic accessibility assessment tools | 55 | 52.4 | 19 | 36.5 |
| Check compliance according to accessibility guidelines | 60 | 57.1 | 25 | 48.1 |
| HTML validation | 47 | 44.8 | 16 | 30.8 |
| CSS validation | 40 | 38.1 | 15 | 28.8 |
| Test with assistive technologies | 32 | 30.5 | 14 | 26.9 |
| Other | 5 | 4.8 | 3 | 5.8 |
| I don't evaluate my designs | 7 | 6.7 | 9 | 17.3 |

## 4.4 Motivations

### 4.4.1 Perceptions of usability for seniors in organizations

Participants were asked to rate the importance given to accessibility for seniors by their organizations or independent practices. While there were varied responses to the question, accessibility for seniors was deemed to be less important for many organizations (31%, n=40). The distribution of the other responses by participants were as follows: 12% very important, 18% fairly important, 15% important, 15% not important.

### 4.4.2 Motivations for usability for seniors

Participants were asked about their organizational and personal motivations in ensuring usability for seniors through two separate multiple selection questions. The most cited motivational factor for organizations was customer requirements (80%), followed by being inclusive (69%) and abiding by the laws (66%). Concerning personal motivations, most of the participants stated being inclusive (82%), followed by being ethical (78%) and developing better products (76%), to be the primary motivations for ensuring usability for seniors (Table 5).

Table 5: Motivations for ensuring usability for seniors

| Motivations | Organizational n | Organizational % | Personal n | Personal % |
|---|---|---|---|---|
| Abiding by the laws | 86 | 66.2% | 60 | 46.2% |
| Being ethical | 75 | 57.7% | 101 | 77.7% |
| Being inclusive | 89 | 68.5% | 107 | 82.3% |
| Customer requirements | 103 | 79.2% | 75 | 57.7% |
| Developing better products | 78 | 60% | 99 | 76.2% |
| Finding research opportunities | 39 | 30% | 50 | 38.5% |
| Increasing income | 55 | 42.3% | 40 | 30.8% |
| Organizational requirements | 55 | 42.3% | 35 | 26.9% |
| Search engine optimization | 24 | 18.5% | 18 | 13.8% |
| Other | 4 | 3.1% | 2 | 1.5% |
| Not sure | 1 | 0.8% | 1 | 0.8% |

## 4.5 Challenges

### 4.5.1 Challenges of ensuring usability for seniors

All participants were asked what the challenges of making websites or apps senior-friendly were through a multiple selection question. The most cited challenges by participants were lack of awareness regarding accessibility for seniors (75%), lack of training/knowledge (74%), time constraints (62%), budget restrictions (60%), and accessibility for seniors not being a requirement for the organization (Table 6).

Table 6: Challenges of ensuring usability for seniors

| Challenges | n | % |
|---|---|---|
| Lack of awareness regarding accessibility for seniors | 98 | 75.4 |
| Lack of training/knowledge | 96 | 73.8 |
| Time restrictions | 81 | 62.3 |
| Budget restrictions | 78 | 60 |
| Accessibility for seniors is not a requirement for the organization | 77 | 59.2 |
| Lack of senior-friendly design guidelines | 74 | 56.9 |
| Accessibility for seniors is not a requirement for the target group/customers | 72 | 55.4 |
| Lack of support from management | 63 | 48.5 |
| Lack of human resources | 41 | 31.5 |
| No legal repercussions | 41 | 31.5 |
| Accessibility for seniors is not seen as a personal responsibility | 33 | 25.4 |
| Accessibility for seniors is outside the job description | 26 | 20 |
| Other | 4 | 3.1 |

## 5 DISCUSSION

This section revisits the findings from the survey and discusses key themes regarding challenges that affect the design of senior-friendly interfaces. The research questions asked in this study were exploratory in nature and were aimed at bringing to light the current practices of UX professionals in the context of designing for seniors. Formulating hypotheses was, therefore, not suitable for the type of research questions asked.

The key contribution of our study is the quantitative data from the survey which we presented in the previous section and interpret here in more detail. In addition to such quantitative data, we are using statements from participants to reflect on the interpretation of the data, which we are bringing into the discussion here. We have not reported the qualitative survey data in the Results section since most of our data was from quantitative surveys, with the free-text answers providing only a small addition to this. These answers were subject to thematic analysis, with the insights gained from this providing nuance and interpretation to the main results.

## 5.1 General Understanding and Awareness

Although the survey was focused on senior-friendly design practices, the results suggest some parallels and connections to web accessibility frameworks which are worth discussing. Various trainings are provided on web accessibility in both industry and academia for design professionals to develop a practical understanding of the accessibility legislation, standards, and guidelines. It is evident from the responses that one-fourth of the participants did not undergo any accessibility training, and although it is concerning, these numbers have improved a lot over the years as evident from previous studies [13][16], which imply that web accessibility training has gained more popularity over time and more professionals are able to access these programs. This distribution of attendance in digital accessibility training was found to be similar to other recent studies on UX professionals in Turkey [19] and the Nordic countries [20].

Regarding familiarity with accessibility legislation, half the participants were not familiar with any government laws on web accessibility. In contrast, most participants in Lazar et al. [24] (74%) were familiar with accessibility legislation. This is important to consider since one of the most important factors influencing organizations to prioritize accessibility is governments enforcing legal compliance with accessibility standards [19][20][24]. On the other hand, in line with previous research [16][19][24], participants were mostly familiar with accessibility guidelines from the Web Accessibility Initiative (WAI), with WCAG being the most well-known set of guidelines and ATAG or UAAG being the least-known. Most participants were also aware of automated accessibility tools similar to Lazar et al. [24]. The level of awareness of accessibility guidelines and tools reported by participants in this study was higher compared to Inal et al.'s study of UX professionals in the Nordic countries [20].

Although participants were generally familiar with different aspects of accessibility, there was a notable lack of awareness among participants regarding designing for seniors. A large number of participants were also not familiar with the senior-friendly design guidelines published by the National Institute on Aging (NIA) and the National Library of Medicine (NLM), which are the most cited set of design guidelines accommodating older adults' needs. Most participants were not aware of other senior-friendly guidelines either, which raises questions and contributes to the discussion regarding the transferability of HCI research-based recommendations from academia to practitioners in the technology design industry [34].

## 5.2 Practical Experiences

Findings regarding current practices of UX professionals reveal that most participants reported considering digital accessibility in the design projects they were involved in, which shows a greater rate of adoption compared to previous studies [13][16][19][24]. This could possibly be a resulting factor of their increased awareness of web accessibility guidelines and tools. Most of the reasons specified by participants for not considering accessibility in their projects (e.g. project scope not including accessibility, target group/customers not requiring accessibility, time and budget constraints, etc.) have also been observed in other studies [19][20]. However, lack of awareness regarding accessibility was not considered to be a reason for participants, unlike previous research [19], where it played a significant role in the non-consideration of accessibility in projects.

In terms of incorporating senior-friendliness, 60% of the participants did not consider usability for seniors in their projects. The most prominent reason behind the lack of consideration of senior-friendliness in their work was that seniors were not their target demographic. In comparison to their consideration of

digital accessibility, while there are a few overlaps in the reasons especially in terms of project characteristics, what stands out are the reasons related to their awareness or expertise in terms of designing for seniors which did not seem to be an issue in the case of accessibility. This is also supported by earlier findings on general awareness (see 5.1), where participants were observed to be more familiar with accessibility compared to usability for seniors.

On comparing the HCI methods used for designing for people with disabilities and those used for designing for seniors, it was found that participants mostly followed an accessibility guidelines-based approach for both demographics. The most common method applied to ensure their design met the requirements of users with disabilities was adhering to accessibility guidelines, followed by conducting heuristic evaluations. It is worth noting here that both these methodologies do not involve the target users and can be conducted without their participation. When designing for seniors, participants again primarily focused on accessibility guidelines, followed by usability tests with seniors, heuristic evaluations, and persona generation based on seniors. In this case, participants considered involving target users to some extent through usability testing, but still focused majorly on HCI methods that did not require user involvement.

Given the high preference for accessibility guidelines, the most common evaluation technique for accessibility among participants was to check for compliance with the said guidelines followed by testing with automated accessibility assessment tools and HTML validation. The same methodologies have also been observed in other studies on UX professionals [19][20]. Only 7% of the participants reported not evaluating their designs for accessibility, compared to 48% in older studies [16], which again shows the increase in accessibility practices adoption in the industry.

Regarding evaluating designs for senior-friendliness, usability testing was the most common technique used to ensure their designs met the needs of senior users, followed by checking for compliance with accessibility guidelines and testing with automated accessibility assessment tools. Usability principles specific to seniors were barely used in the design of user interfaces for older adults, and this could be attributed to the earlier finding regarding the lack of familiarity with senior-friendly design guidelines (see 5.1).

## 5.3   Motivations and Challenges

UX professionals' motivations for ensuring usability for seniors and the challenges they face in the process were examined through the following dimensions: perceptions of usability for seniors in organizations, motivations for usability for seniors at the organizational level and at an individual level, and challenges of ensuring usability for seniors.

Most organizations represented in this study deemed usability for seniors to be 'less important', in contrast to Inal et al.'s [20] findings on organizational perspectives, where digital accessibility was perceived to be an important asset to many organizations. The main drivers to ensure usability for seniors for these organizations were customer requirements, inclusion of all users, and legal repercussions. Participants believed that their organizations would be more interested in ensuring usability for seniors if it was required by their customers. They also thought that their organizations would be motivated to incorporate senior-friendliness if they realized the need to be inclusive to all user groups and if they were obligated by law. These findings are similar to Lazar et al. [24], where government regulations and knowing that people with disabilities are using their websites were the biggest motivators for participants to make their websites accessible, and can be observed in other more recent studies as well [16][19][20]. From a personal perspective, inclusivity, ethics, and the desire to develop better products were reported to be the main drivers for taking usability for seniors into account. The concept of ethics was also discussed by Lazar et al. [24] as most participants in their study reportedly considered ethics to be important in the development of accessible websites.

Regarding challenges of ensuring usability for seniors, the most important challenges stated by the participants were lack of awareness regarding accessibility for seniors, lack of training or knowledge, time and budget restrictions, and accessibility for seniors not being a requirement for the organizations. Other challenges cited by participants, in descending order of frequency, include lack of support from management, lack of human resources, no legal repercussions, accessibility for seniors not being seen as a personal responsibility, and accessibility for seniors being outside the job description. Some of the key themes that emerged from participants' responses regarding challenges that affect the design of senior-friendly interfaces are discussed below:

### 5.3.1   Seniors are not the target users

Generally, the design requirements of products and services are based on the needs and pain points of the target user group. Based on responses from the participants, it is evident that seniors are barely considered as part of the main target demographic, even for applications that are generic in nature. Our recruitment of professional designers for the survey was broad and we did not limit our focus to particular applications. Although some designers may primarily work on applications or websites that are not intended for older adults (e.g. children's apps), none of the survey respondents mentioned this fact in the background information or any other free-form text about their design activities.

One of the main reasons behind the non-consideration of older adults in the design process is the common misconception that seniors are not tech-savvy or they are not using such online services. As a result, designing for them is often overlooked in favor of target user groups that are perceived to be more profitable, thus contributing to "digital ageism". Complementing several market and government census reports, research data from across the globe show that the percentage of older adults that use the Internet is increasing [14][32][35][39]. Due to their perceived lack of senior users, many organizations are losing out on customers by not putting in the required effort to meet the needs of a considerable segment of their audience.

### 5.3.2   Lack of standardized senior-friendly design guidelines

Another challenge mentioned by participants was the lack of design guidelines that focused specifically on the needs of senior users. This was expected as very few design professionals were familiar with the guidelines published by NIA and NLM, or other guidelines. Of the 52 participants who reported considering usability for seniors in their projects, only 8 were familiar with these guidelines. This implies that these guidelines are barely used when designing for seniors. It is also evident from responses to other questions in the survey that participants were more familiar with the web accessibility guidelines and preferred using them, as opposed to the senior-friendly design guidelines, when designing for seniors. This lack of familiarity with senior-friendly guidelines can be attributed to the fact that they are not as universal or standardized as the web accessibility guidelines.

### 5.3.3 Lack of support from stakeholders

Another common barrier to senior-friendly design as cited by participants was the lack of support from stakeholders or clients who commissioned the designers' services. Most clients are not aware nor knowledgeable about the need for senior-friendly designs, and as a result, the project briefs provided by them barely include accessibility for seniors as a crucial requirement. In order to consider accessibility for seniors in projects, UX professionals need additional time and resources, although budgets for these processes are often too restricted. Unless the client is on board, it is difficult for UX professionals to get the budget or the time to incorporate the needs of senior users, or to convince them why certain design choices must be made to accommodate related concerns. One participant stated:

> "Once the client realizes this is a target market, there is no longer a question about UX for seniors. It all begins with the client."

If usability for seniors is not listed as a client requirement, it comes down to the time and cost budgeted for the project, and then accessibility for seniors is no longer a priority.

### 5.3.4 Aesthetics vs accessibility

An important aspect that was brought up by a few participants was the prioritization of aesthetics over accessibility for seniors. Participants mentioned that the stakeholders did not care much about accessibility because the elegant design is what attracted new business, as also evidenced from Lazar et al. [24]. As a result, they would rarely budget for accessibility. Many designers also had a similar approach to this, assuming that in order to design for seniors, the trade-off would be a generic, less attractive, and less engaging product. For example, one participant mentioned:

> "Sometimes we let design overrule contrast warnings and text size warnings since these don't affect the vast majority of our non-senior, non-consumer audience".

However, as evident from previous studies [48], when user interfaces are designed to be accessible, they render a positive user experience for both users with and without disabilities.

## 6 KEY INSIGHTS

This study highlighted several key issues that UX professionals face with respect to making their products more usable and more accessible to seniors. A summary of these issues has been included below. Uncovering these, in our view, is an essential step toward addressing the lack of senior-centered focus within the UX practice. Some of these insights are similar to those exposed by Lazar et al. [24] with respect to accessibility, which suggest that, (a) designing for seniors is yet to "catch up" to the gains made with respect to designing for accessibility, and (b) the issues uncovered here are not intractable, as Lazar et al.'s work [24] acted as the spark for numerous changes in accessible design. Further research is needed to determine the appropriate course of action to address the issues and gaps that our study exposed (which is outside of the scope of this paper and would be too speculative to include here). Meanwhile, we invite the broader research and design practice community to use these as starting points in reflecting on approaches to address the many issues identified by our survey.

1.  While UX professionals are generally aware of web accessibility guidelines, tools, and assistive technologies, their level of awareness regarding how to design for seniors and the availability of senior-friendly design principles is notably low.
2.  Very few UX professionals consider usability for seniors in the design projects they are involved in, primarily due to senior-friendliness not being a requirement of the target user group and lack of knowledge regarding designing for seniors.
3.  The main methodologies used by UX professionals when designing for senior users are to follow accessibility guidelines and to conduct usability tests with older adults.
4.  The familiarity with, and the use of senior-focused usability principles among UX professionals is minimal despite the availability of a wide variety of research-based recommendations.
5.  Organizations are motivated to ensure usability for seniors in their products when their customers require it, when they want to be inclusive to all user groups, and when it is required by law.
6.  At a personal level, UX professionals are motivated to design for seniors due to inclusiveness, ethics, and the desire to develop better products.
7.  Older adults are generally not considered to be the target demographic by most organizations, which leads to stakeholders not budgeting for the time and resources required to ensure usability for seniors.
8.  Higher emphasis is placed on visual design and aesthetics compared to accessibility features and usability needs for seniors.

## 7 LIMITATIONS AND FUTURE WORK

While our study draws methodologically from prior research, including following similar sample sizes, and using validated instruments, there are inherent limitations to our findings. Primarily, these limitations come from the exclusive use of Internet-based surveys – the only research method available to us during significant periods of pandemic-related lockdowns and restrictions to research activities. In coordination with our university's ethics and research office we have implemented various mechanisms to ensure that survey responses are completed in good faith; however, these mechanisms are not able to verify the specific accuracy of responses (e.g. time spent in industry, or number of projects worked on).

There are additional limitations inherent to surveys as a data collection method, such as not answering "Why" questions and gaining a deeper understanding of the respondents' challenges they face in their design practice. We plan to conduct follow-up in-person contextual inquiry sessions with some of our survey's respondents (most have provided us with their contact for follow-up), which will be situated in the context of their work or practice.

## 8 CONCLUSION

This research focused on investigating the perspectives and practices of design professionals in the context of designing for seniors. The study was conducted using an online survey, and 130 design professionals from various industries participated in this research. The results of the study show that most UX professionals are familiar with web accessibility guidelines and assistive technologies. However, there is a considerable lack of awareness regarding how to design for seniors, and a large number of design professionals are also not familiar with any senior-friendly design guidelines. Results also suggest that only few UX professionals consider usability for seniors in the design projects they are involved in. The primary reasons cited for this are senior-friendliness not being a requirement for the target group/customer, lack of awareness of how seniors use the Internet, senior-friendliness not required by clients/stakeholders, and senior-friendliness not being a priority.

This study opens the door for future investigations that may explore and validate approaches to improving UX professionals'

awareness of designing for seniors. A follow-up study will focus on larger scale surveys that refine our understanding gained in this research, and which will allow for more complex factor analysis. Further research will also include in-person contextual inquiry sessions with participants. The primary goal of this study was to bring to light the lack of awareness and understanding that UX professionals have in terms of designing for seniors, and to identify some of the very specific causes of this issue. The knowledge obtained about these causes is a first, and a very important step toward addressing the overarching lack of consideration of seniors in the design of user interfaces. Similar to Lazar et al. [24], this study lays the groundwork for other researchers to propose ways to address this issue and improve the state of usability for seniors in the UX practice. Overall, it is a valuable account of the current state of awareness and activity in the field of technology design with regards to usability for older adults, and a reminder that there is much work to be done to promote the how and why of designing for an older audience.

### ACKNOWLEDGEMENTS

The authors wish to acknowledge the support received from AGE-WELL, Canada's technology and aging network federally-funded through the Networks of Centres of Excellence (NCE) program. Dr. Cosmin Munteanu also acknowledges the support provided through the Schlegel Research Chairship in Technology for Healthy Aging at the Schlegel-UW Research Institute for Aging.

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
