# OpenReview forum: "“Not Our Target Users”: UX Professionals’ Perceptions of Designing for Older Adults"
_graphicsinterface.org/Graphics_Interface/2023/Conference_SD — GI 2023 - second deadline_

### Official Review · Reviewer_xv1t · 2023-04-10
**In this paper, the authors present a perspective that builds off of an existing body of work surrounding accessibility research. They apply existing methods from accessibility research towards designing for older adults in order to uncover challenges or barriers that exist towards this practice.**

**Rating:** 6
**Confidence:** 3

**Review:**

In this paper, the authors build upon an existing body of work in order to contribute insights towards the current state of how design professionals consider digital accessibility and usability for the ageing population when it comes to UX design. They conducted an online survey with 130 professionals working in UX roles in order to understand the level of understanding and awareness towards accessibility and usability for seniors, how UX professional incorporate such principles into their designs for seniors, and what motivations, barriers, and challenges exist when creating senior-focused designs. The authors present a set of findings that highlight a lack of knowledge and awareness regarding approaches for making technologies more accessible, and the usability needs of older adults as the main barriers to senior friendly design.

Pros/Positives:
- The authors argue that accessibility is widely explored when it comes to practices for people with disabilities, however, there is a lack of focus when it comes to professionals' expertise and experience designing for seniors or older adults. This, along with other motivation for the work is backed by literature in HCI, demonstrating the timeliness and relevance of this work in the current research space and state of being.
- The methodology is clearly articulated, including reference to previous works whose methods were followed or inspired this project, as well as an in-depth explanation of what the online survey looked like and consisted of. The authors do a good job of walking through specific details of the types of questions that were included, where they originated from, and how they were extended to be more relevant for their purposes.
- The narrative throughout the paper is clear - i.e., the motivations and goals stated in the introduction connect directly to the research questions that are explicitly stated in the methods, and the results are clearly derived to answer each of these. The discussion further ties these findings back to related literature and previous knowledge.
- The concluding "Key Insights" sections nicely summarizes the study's findings and contributions.

Questions/Concerns:
- In the introduction, multiple references are made to things such as "design needs", "needs of the ageing population", and "senior-friendly" design. However, these specific needs and what exactly qualifies within these categories are not expanded upon, leaving the reader to wonder what exactly might be required to increase accessibility and usability for older adults as compared to their younger counterparts?
- In the results section, it is a bit unclear what is the difference between designing for users with disabilities and designing for seniors as it relates to accessibility (based on survey responses), except for when considering evaluation approaches.
- I am also a little bit confused as to why questions regarding motivations were posed as "… by their organizations or independent processes." Were the two aligned in all cases? I can imagine there would be instances where someone cared more (or less) about ensuring usability and accessibility for seniors when compared to their organization. Were such cases supported by the survey structure and setup? How so?
- The discussion seems to beat around implications for HCI researchers and the UX community that arise from their findings, and presents a large amount of repetition towards the results sections (especially in 5.1 and 5.2).

---

### Official Review · Reviewer_Z9mv · 2023-04-20
**Messy but useful paper**

**Rating:** 6
**Confidence:** 3

**Review:**

This paper presents the results of a survey of 130 UX professionals to gauge their consideration of the needs of older adults in their design process as well as their knowledge of and adherence to guidelines for making websites accessible for older adults. The work contributes empirical evidence demonstrating low awareness and application of guidelines.

First, I applaud the authors for tackling this very complex subject. The work is timely and vital. As the work demonstrates, older adults are often overlooked in the technology design process. Concrete steps (such as this) are needed to understand the challenges and barriers to more age-inclusive technology design.

However, while I was delighted to see the paper acknowledge that designing for older adults is not the same as ensuring accessibility, I was disappointed not to see this deeply mapped out. There is little acknowledgment of how ‘usefulness’ factors into designs for later life. Systems are typically designed around the envisioned needs of a young persona which can (often!) lead to feature designs and prioritization that are not well-matched to the needs and preferences of older adults, which makes these systems less useful for older adults and less easy to use. Instead, the paper often seems to classify design for older adults as a sub-type of accessibility. This can be subtle (e.g., in the discussion, accessibility awareness is discussed first and then aging in a way that seems to cast it as a subtype). There are also places like Table 6 where the scoping is framed as “accessibility for seniors” (while the table itself is labelled usability for seniors), further hindering interpretation.

I had a hard time following section 2.1 and am concerned it could misrepresent the distinction between usability and accessibility and the definition of disability. W3.org (https://www.w3.org/WAI/fundamentals/accessibility-usability-inclusion/) discusses these concepts and notes (much more than the discussion in 2.1) the substantial overlap and the social constructed-ness of the distinction. This is particularly clear in the sentence describing how the ISO definition of usability “could address accessibility when: ‘specified users’ includes people with a range of disabilities, and ‘specified context of use’ includes accessibility considerations such as assistive technologies. But usability practice and research often does not consider the needs of people with disabilities.” A phrase in this section also suggests that older adults are not disabled because their accessibility barriers are gradual and/or due to ageing. I think there is some nuance that is missed here. Perhaps the intention was that many older adults do not consider themselves disabled because their change in ability is an expected part of the aging process. It might seem like I’m being nitpicky here, and to researchers working in this area, it might not matter if there is a bit of imprecision in this section. But if this paper aims to raise awareness of the need for more attention to older adults in design (and I think it is!), then it is imperative that those unfamiliar with these distinctions can precisely follow.

One significant nuance to designing for older adults is addressing diversity. The paper quite appropriately notes that age alone doesn’t define the group, but it doesn’t go further or try to capture that diversity in the survey. As such, we are left not really knowing what is meant by designing for older adults. This is important because what it means to consider the needs of older adults in the design process could vary substantially depending on the system and context for use. A system deployed in a business setting where the only older adult users are of the youngest old (still in the workforce) and are highly trained/experienced might not need to consider older adults' needs (beyond general accessibility). A system designed for teenagers might not need to consider older adults at all. A system designed for a general population, including the oldest old, would require much more explicit design attention. The paper notes, “60% of the participants did not consider usability for seniors in their projects”. I’m sure most of these situations would have benefitted from explicit consideration of older adult needs, but likely some did not. That this is not discussed in the paper limits the interpretability of the results and risks some readers will over-emphasize the size of the ‘didn’t really need to consider older adults’ category.

A further difficulty with the survey is the statement \“I know how seniors use websites.” I’m not sure that this is an answerable question. If it is, the answer would be highly nuanced and complicated. I’m not sure anybody could honestly answer yes to this (Certainly, I’m not comfortable with answering yes despite being a researcher in this area!) I’m not sure if the idea would be that people would answer yes if they were familiar with some of the accessibility workarounds older adults use, the accessibility features they would tend to benefit from, or the challenges they encounter. Still, none of these really match up with the statement’s phrasing. I would like to see the survey design acknowledged as an initial attempt and its flaws outlined.

As a more minor point, the paper uses the word ‘senior’ throughout while noting that ‘older adult’ is more commonly used in HCI. I don’t think the term ‘senior’ is offensive, but ‘older adult’ is preferred for good reasons, notably clarity (seniors can refer to the most senior in various contexts and not just for age, e.g., high school seniors). I’m not convinced by the rationale provided that “the more prevalent term in our own [Canadian] sociocultural context is that of ‘senior’.” Stats Canada prefers the term older adult, specifically noting language precision as the reason (https://www.statcan.gc.ca/en/subjects-start/older_adults_and_population_aging). I can see an argument for using the word seniors in the survey instrument (I think there is a reasonable argument to be made that this word is more common in lay usage), but not for the paper.

Overall, I lean slightly towards accepting this paper. As outlined above, I think there are many weaknesses with how the research was designed and how the paper is presented. And yet, I don’t think these weaknesses result from sloppiness. Instead, I think these are very complicated, nuanced discussions and expecting them right on the first try is perhaps unrealistic. Accordingly, I think there is a benefit in publishing this so that others can build on it. It would, however, be helpful if the limitations could be better acknowledged in the paper to support future efforts.

---

### Official Review · Reviewer_JUwG · 2023-05-02
**Good paper**

**Rating:** 7
**Confidence:** 3

**Review:**

This paper present an important, timely, well thought revisitation of Lazar's work on understanding designers’ perceptions of accessibility for people with disabilities, except this time for older adults / seniors.

The work is very well motivated and grounded in the literature review, that does a great job at outlining what is missing and how the proposed research addresses existing gaps.

The paper is well written and easy to follow; the methodology described in details and the instruments used well explained as well.

I find the restults section and most of the discussion (Sections 5.1 and 5.2) to be very redundant. I would recommend either merging these, or making 5.1 and 5.2 much shorter; so that the discussion can focus on the most new/interesting parts that are in 5.3.

I am a bit confused about section 5.3.1. The authors seem to assume that it is necessarily a bad thing if professionals think that seniors are not their target users. But there are definitely plenty of cases where seniors are indeed not the target users, and considering them might not be appropriate (e.g., a game or educational tool for children). Although this seems obvious, I think this should be discussed more, because the results from the survey can lead to confusion on that point.

The formatting of multiple citations in the paper should be fixed ([1,2,3] instead of [1],[2],[3]).
The limitations and future work header uses the wrong formatting.